# Dysfunctions of Neutrophils in the Peripheral Blood of Children with Cystic Fibrosis

**DOI:** 10.3390/biomedicines11061725

**Published:** 2023-06-15

**Authors:** Ganimeta Bakalović, Dejan Bokonjić, Dušan Mihajlović, Miodrag Čolić, Vanja Mališ, Marija Drakul, Sergej Tomić, Ivan Jojić, Sara Rakočević, Darinka Popović, Ljiljana Kozić, Miloš Vasiljević, Marina Bekić, Srđan Mašić, Olivera Ljuboja

**Affiliations:** 1Pediatric Clinic, Clinical Center of the University of Sarajevo, 71000 Sarajevo, Bosnia and Herzegovina; ganimeta.bakalovic@gmail.com; 2Center for Biomedical Sciences, Faculty of Medicine Foča, University of East Sarajevo, 73300 Foča, Bosnia and Herzegovina; dejan.bokonjic@ues.rs.ba (D.B.); dusan.a.mihajlovic@gmail.com (D.M.); miocolic@gmail.com (M.Č.); vanjamalis916@gmail.com (V.M.); marijadrakul@gmail.com (M.D.); jojke@yandex.com (I.J.); saradrakocevic@gmail.com (S.R.); darinkadubovina91@gmail.com (D.P.); ljiljanakozic8@gmail.com (L.K.); vasiljevicmilos85@gmail.com (M.V.); smasic@gmail.com (S.M.); 3Department of Pediatrics, Faculty of Medicine Foča, University of East Sarajevo, 73300 Foča, Bosnia and Herzegovina; 4Serbian Academy of Sciences and Arts, 11000 Belgrade, Serbia; 5Institute for the Application of Nuclear Energy, University of Belgrade, 11080 Belgrade, Serbia; sergej.tomic@inep.co.rs (S.T.); marina.bekic@inep.co.rs (M.B.); 6Clinic for Children’s Diseases, University Clinical Center of Banja Luka, 51000 Banja Luka, Bosnia and Herzegovina

**Keywords:** cystic fibrosis, children, neutrophil functions, cytokines, flow cytometry

## Abstract

Dysfunction of neutrophils in patients with cystic fibrosis (CF) is best characterized in bronchoalveolar lavage (BAL), whereas peripheral blood neutrophils are less examined, and the results are contradictory, especially in younger populations. Therefore, this work aimed to study functional and phenotypic changes in circulating neutrophils in children with CF. The study included 19 CF children (5–17 years) and 14 corresponding age-matched healthy children. Isolated neutrophils were cultured either alone or with different stimuli. Several functions were studied: apoptosis, NET-osis, phagocytosis, and production of reactive oxygen species (ROS), neutrophil elastase (NE), and 11 cytokines. In addition, the expression of 20 molecules involved in different functions of neutrophils was evaluated by using flow cytometry. CF neutrophils showed reduced apoptosis and lower production of NE and IL-18 compared to the healthy controls, whereas IL-8 was augmented. All of these functions were further potentiated after neutrophil stimulation, which included higher ROS production and the up-regulation of CD11b and IL-10 expression. NET-osis was higher only when neutrophils from moderate–severe CF were treated with *Pseudomonas aeruginosa,* and the process correlated with forced expiratory volume in the first second (FEV1). Phagocytosis was not significantly changed. In conclusion, circulating neutrophils from children with CF showed fewer impaired changes in phenotype than in function. Functional abnormalities, which were already present at the baseline levels in neutrophils, depended on the type of stimuli that mimicked different activation states of these cells at the site of infection.

## 1. Introduction

Cystic fibrosis (CF) is the most common genetic disease in Caucasians and is the result of a mutation in a gene encoding the cystic fibrosis transmembrane conductance regulator (*CFTR*) [1,2]. This gene encodes for the CFTR protein, which is responsible for the passive transport of chloride (Cl^−^) and bicarbonate (HCO^3−^) across epithelial membranes [1,2]. Lung disease, which is characterized by chronic inflammation, progressive airflow obstruction due to dehydration, hypersecretion of mucus, and subsequent bacterial infections of the airway, is a significant cause of morbidity and mortality in patients with CF due to impaired ion transport [3]. In addition, the exaggerated inflammation, in synergy with its impaired resolution—dominated by infiltrated neutrophils—leads to progressive lung damage with bronchiectasis formation, wall thickening, and many functional defects [1].

In CF, neutrophils represent the first line of defense and, thus, play a vital role in anti-microbial activity. Neutrophils become activated through interaction with microbial pathogens, proinflammatory cytokines, and sterile inflammatory agents [4]. Activated neutrophils kill microorganisms through phagocytosis, by forming neutrophil extracellular traps (NETs), and through degranulation. These innate immune cells also act indirectly through the release of proinflammatory cytokines, such as tumor necrosis factor-αlfa (TNF-α), interleukin (IL)-1β, and chemokines, including IL-8 (CXCL8), and monocyte chemoattracting protein-1 (MCP-1) (CCL2), as well as other cytokines with regulatory functions [5]. These cytokines/chemokines stimulate the production of other biomolecules that recruit more innate immune cells to the site of infection or inflammation [6]. However, many results suggest that CFTR dysfunction causes a dysregulated inflammatory response, which is partly due to defective expression of CFTR in immune cells, and that CF airways are already a perfect microenvironment for amplifying the immune–inflammatory mechanisms before any infection [1,7]. 

Neutrophils have an active pro-inflammatory role in CF lung disease and play a regulatory role with the possibility of high adaptation, reprogramming, and plasticity [8]. Up to now, significant changes in the physiology and phenotype of neutrophils have been detected in CF patients’ bronchoalveolar lavage (BAL). Airway neutrophils of CF patients showed a blunted phagocytic capacity, reduced expression of cell surface recognition receptors, such as Toll-like receptors (TLRs), and reduced reactive oxygen species (ROS) production, leading to impaired bacterial killing [9,10]. In the airways, CF neutrophils undergo active exocytosis of primary granules and NET-osis, leading to a massive release of enzymes, such as neutrophil elastase (NE) and myeloperoxidase (MPO), and DNA, thus damaging the airway tissue and perpetuating inflammation [11]. Very often, these abnormalities can be fatal in CF patients. 

However, alterations of some functions of neutrophils in peripheral blood have also been reported. Interest in studying circulating neutrophils in CF patients increased significantly after discovering that CTFR was also detected in neutrophils [12]. Although the expression of CTFR at the membrane of phagolysosomes in neutrophils is relatively low, several clinical and experimental studies demonstrated that the absence or dysfunction of CFTR in neutrophils impairs bacterial killing [13,14,15]. Phagolysosomal CFTR was shown to be essential for chloride transport and was linked to bacterial protein chlorination during phagocytosis. Therefore, the absence of CFTR in neutrophils results in a defect in intraphagolysosomal HOCl production [15], increased cytosolic chloride and sodium levels, decreased magnesium levels, and an impaired degranulation of secondary and tertiary granules [16]. Importantly, this neutrophil defect was partially restored in patients with the G551D mutation by the ion channel potentiator ivacaftor [16,17].

Despite these intriguing findings, circulating neutrophils from CF patients showed impaired chemotaxis [18], production of IL-8 and IL-1 receptor antagonist (IL-1Rα) [19], and oxidative burst [15,17,20,21]. In addition, blood CF neutrophils had decreased capacity for killing *Pseudomonas* (*P.*) *aeruginosa* in vitro [1,22]. Dysfunction of circulating neutrophils might also be related to the low-level endotoxemia originating from a chronic lung infection, independently of CFTR defects, or due to increased production of pro-inflammatory cytokines and chemokines, especially in patients with exacerbation of the disease [23]. Therefore, although the relationship between abnormal CFTR function and the development of airway inflammation has been suggested as a primary event in CF, the pathogenic role of neutrophils has not yet been fully elucidated. In this context, it is unclear whether nonfunctional CFTR leads to primary neutrophil changes before migration into the lung parenchyma or whether neutrophils become dysfunctional in the airway microenvironment [24]. If the first hypothesis is correct, then blood sampling is a more straightforward and practical method for the isolation of neutrophils than their collection from the lower respiratory tract, especially in young children. Although BAL provides an adequate picture of the inflammation in children with CF, performing bronchoscopy can be risky. Keeping these facts in mind, our primary aim was to examine cell surface marker expression, production of ROS, NE, and cytokines, antibacterial function, phagocytosis, apoptosis, and NET-osis of peripheral blood neutrophils in children with CF—predominantly those with normal and mildly changed lung function—and to compare these parameters with those of neutrophils from healthy children. Such a complex study has not been previously published. 

## 2. Materials and Methods

### 2.1. Study Participants

This was a cross-sectional clinical study with the predominance of ex vivo laboratory testing on *n* = 33 participants from Bosnia and Herzegovina (BiH) who were recruited during the period of June 2018–October 2020. Children and adolescents aged 5 to 17 years (*n* = 19) from Bosnia and Herzegovina (BiH) with a diagnosis of CF who were being treated at the Pediatric Clinic of the Clinical Center of the University of Sarajevo (CCUS) in the Federation of BiH, and the Clinic for Children’s Diseases of the University Clinical Center of Banja Luka in the Republic of Srpska (RS) were included in the study. Of them, ten were male, and nine were female. The participants comprised 43% of the pediatric CF population in BiH (*n* = 44), as there were 50 registered cases in the country. The main inclusion criterion was the absence of any sign of acute infection during clinical examination and laboratory testing. The diagnosis of CF was established previously according to the Consensus Guidelines for CF [25], and it included the determination of chloride concentration in sweat (sweat test) by using the standard pilocarpine iontophoresis technique. Values over 60 mmol/L were considered positive [25]. The detection of two mutations in the *CFTR* gene in all patients was performed on the 30 most common mutations. The testing was performed at CCUS (patients from the Federation of BiH) and at the Institute for Maternal and Child Health Care, Republic of Serbia (patients from RS). Our study group consisted of 84.21% carriers of the F508del/F508del mutation, and 15.79% corresponded to compound heterozygotes (F508del/R10070Q and SR466X; F508del/N1303K; F508del/unknown).

The study, which followed the STROBE guidelines for reporting observational studies [26], was conducted according to the guidelines of the Declaration of Helsinki and approved by the Ethics Committee of the Faculty of Medicine Foča, University of East Sarajevo, BiH (protocol code: 01-5/45, date of approval: 4 May 2018). Signed informed consent was obtained from the parents of the participants.

Nineteen patients with CF were included in the study. The mean age was 12.53 ± 3.89 years (range: 5–17 years). Of them, 12 (63.2%) were male and 7 (36.8%) were female. The control group (*n* = 14) consisted of healthy children of whom routine laboratory analyses were performed as part of regular systematic examinations. This group had normal physical findings, no signs of current acute infection or infection during the last month, and no history of chronic diseases, including allergies. The participants in both groups who had taken oral steroids, antibiotics, or antioxidant supplements in the last month were excluded. The control group had a similar mean age (13.90 ± 3.11) (range: 6–16 years) and a similar sex ratio (8 males, 57.14%; 6 females, 42.86%) to those of the children with CF. The mean body mass index (BMI) of the CF and control groups was 16.50 ± 2.41 kg/m^2^ and 17.42 ± 2.22 kg/m^2^, respectively. The demographic characteristics of the study participants are presented in Table 1. Statistically significant differences between the groups were not detected for any parameters (*p* > 0.05). 

The laboratory part of the study was performed at the Center for Biomedical Sciences, Faculty of Medicine Foča. 

### 2.2. Clinical Examination

The general clinical examination included the general health status (based on an interview, data from individual CF patient files, and clinical measurements of anthropometric data (weight, height, body mass index/BMI)) and spirometric data. Height and body mass were measured in centimeters and kilograms, respectively. The BMI was calculated with the ratio of body mass and height squared and was expressed in kg/m^2^ during data processing. Functional lung tests were performed with a Jaeger Master Scope spirometer (Jaeger GmbH, Hockberg, Germany) by using a standardized procedure. The children did not take short-acting beta-2-agonists for 24 h before the test. The tests included forced expiratory volume in the 1st second (FEV1) and forced vital capacity (FVC). The degree of lung damage was classified in relation to FEV1 into four groups. In Group I, an FEV1 of 90% or more was considered normal. In Group II, an FEV1 from 89% to 70% was characterized as mild lung damage. In Group III, an FEV1 from 69% to 40% had the characteristics of moderate lung damage. In Group IV, an FEV1 below 40% was recognized as severe lung damage [27].

The data collected from the interviews and CF history files included the age of CF diagnosis, the onset of disease symptoms, initial clinical manifestation, molecular genetics profile, the severity of the lung disease, and disease control. The complications of the disease that were recorded included complications of bronchopulmonary aspergillosis, pulmonary hypertension, type I diabetes, chronic colonization with *Burkholderia cepacia,* and *P. aeruginosa*. 

Documented infection meant the presence of clinical symptoms and signs of infection with microbiological confirmation of infection. Microbiologically confirmed infection implied the isolation of the causative agent from bronchoalveolar lavage (BAL), sputum, or a deep pharyngeal swab, and it was inoculated on nutrient media according to the usual laboratory procedure. By using a variety of common cultivation media, as well as special cultivation media (BCSA—*Burkholderia cepacia* selective agar), microorganisms that colonized the respiratory tracts of the CF patients were identified. Chronic colonization with *Staphylococcus aureus* meant the presence of more than two positive cultures during one year, despite prophylactic antibiotic therapy [28]. Chronic colonization with *P. aeruginosa* was defined according to the Leed criterion, which included more than 50% of positive cultures with mucoid strains of *P. aeruginosa* from 12 (minimum 8) cultures taken and analyzed in one year [29].

The diagnosis of allergic bronchopulmonary aspergillosis implied the presence of positive clinical, standard radiographic, or CT presentation in laboratory, serological, or microbiological tests. Confirmation of diagnosis was based on the fulfillment of major or minor criteria [30]. Pulmonary hypertension was diagnosed by using cardiac ultrasonography, which included elevated pressures in the pulmonary artery (systolic pressure in the pulmonary artery above 30 mm Hg, diastolic pressure above 10 mm Hg, and mean pressure above 20 mm Hg), increased pulmonary vascular resistance, and clinical signs of right heart failure [31]. The evaluation of the function of the endocrine pancreas and the diagnosis of DM were performed based on the criteria of the World Health Organization for the diagnosis of diabetes and prediabetes in patients with CF.

The assessment of disease control was performed based on three parameters: FEV1—determined according to the values at the last control examination; the number of exacerbations of lung disease during one year, which required hospitalization; BMI. Based on the mentioned parameters, the patients were classified into three groups: Group I—satisfactory control of the disease (BMI > P25; FEV1 > 60%, at most one hospitalization during the year); Group II—poor disease control (BMI < 60%, more than one hospitalization during the year); Group III—fatal outcome (which was not present in our study).

### 2.3. Isolation of Peripheral Blood Neutrophils 

Peripheral blood was sampled in vacuum-filled tubes with the addition of K_2_EDTA and after a clinical examination of both study groups. Granulocytes were isolated from the sampled blood via sedimentation with a 3% dextran solution. After sedimentation, leukocyte-enriched plasma was isolated and applied to the Lymphoprep gradient (PAA Laboratories, Vienna, Austria) with a density of 1.077 g/mL. After centrifugation (2200 rpm, 20 min at room temperature), granulocytes were isolated and residual erythrocytes were lysed with a lysing buffer (NH_4_Cl + KHCO_3_ + Na_2_EDTA). Granulocytes were resuspended in Hanks’ balanced salt solution (HBSS) medium (Sigma-Aldrich, Steinheim, Germany) without Ca^2+^ and Mg^2+^ and, as such, were used for further experimental purposes. The purity of the cell suspension (usually higher than 95% neutrophils) was determined by staining the cell suspension with Türck’s solution. The viability (usually higher than 95%) was determined by staining the cells with a 0.2% solution of Trypan blue (Sigma-Aldrich, Steinheim, Germany). Cells were counted by using a Neubauer chamber under a light microscope. 

### 2.4. ROS Production

A chemiluminescence assay with luminol was used to detect ROS production. Neutrophil granulocytes that were isolated from the blood of CF patients and healthy children as previously described were resuspended in an HBSS medium with Ca^2+^ and Mg^2+^ enriched with 1% human inactivated serum and 10 mM HEPES (Sigma-Aldrich, Steinheim, Germany) and plated in a 96-well white plate (2.5 × 10^5^/150 µL per well). After incubation (30 min at 37 °C), 50 µL of luminol (Serva, Munich, Germany) in HBSS medium was added to each well. After incubation (15 min at 37 °C), different stimuli were added: phorbol myristate acetate (PMA) (20 nM), calcium ionophore A23187 (CaI) (1 µM), opsonized zymosan (OpZym) (10 µg), *N*-formyl-methionyl-leucyl-phenylalanine (fMLP) (1 µM) + lipopolysaccharide (LPS) (100 nM), and *P. aeruginosa* at a multiplicity of infection of 10 (MOI 10). The simultaneous addition of LPS—as a neutrophile priming agent—and fMLP—as a triggering event—is a standard procedure for inducing ROS production or better stimulation of neutrophils [32]. All chemicals were from Sigma-Aldrich (Steinheim, Germany). *P. aeruginosa*, ATCC 9027 strain, was obtained from the ATCC collection in Rockville, MA, USA. The intensity of emitted light, which was proportional to the production of ROS, was measured immediately after stimulation with a chemiluminescent spectrometer (Synergy HTX, Bio Tek Instruments, Santa Clara, CA, USA). The release of ROS was monitored every 2 min for 3 h at 37 °C, and the values are shown as the area under the curve (AUC).

### 2.5. Apoptosis Assay

Apoptosis of neutrophils was measured by using a commercial apoptosis kit (Biolegend, London, UK) based on cell labeling with Annexin V conjugated with allophycocyanin (APC) and propidium iodide (PI). Neutrophils (2.5 × 10^5^/150 µL per well) were cultured in a plate of 96 wells in HBSS medium with Ca^2+^ and Mg^2+^ enriched with 1% human inactivated serum and 10 mM HEPES for 18 h in an incubator with 5% CO_2_ at 37 °C. Both unstimulated cells and cells stimulated with various stimuli, such as fMLP/LPS, OpZym, and *P. aeruginosa* (MOI 10), were used. After incubation, neutrophils were washed, followed by adding binding buffer from the commercial kit and the fluorescent dyes to the cell pellet as described by the manufacturer. Neutrophils were incubated for 20 min in a dark place, and after washing them in the binding buffer, the samples were analyzed with a flow cytometer (Attune, Thermo Fisher Scientific, Waltham, MA, USA). Annexin V-APC^+^ and Annexin V-APC^+^/PI^+^ cells were recognized as apoptotic cells. The results were expressed as % apoptotic neutrophils.

### 2.6. Phenotypic Analysis of Neutrophils with Flow Cytometry

The phenotypic characteristics of in vitro cultured neutrophils were determined with direct immunofluorescence by using a panel of monoclonal antibodies (Abs) that were characteristic for neutrophils. The names of the conjugated Abs, dilutions, and manufacturers are given in Appendix A. The manufacturers were Biolegend (San Diego, CA, USA), Thermo Fisher Scientific (Waltham, MA, USA), and R&D Systems (Minneapolis, MN, USA). Neutrophils were cultured under conditions identical to those described for the apoptosis assay. After 18 h of cultivation, the cells were washed and resuspended in cold phosphate-buffered saline (PBS) with 0.01% sodium azide at a concentration of 1 × 10^5^ cells/50 μL. Blocking antibodies for the Fc receptor were first added to the cell suspensions, followed by monoclonal antibodies in final dilutions according to the manufacturer’s recommendations. The cells were incubated for 30 min at +4 °C, washed, and then analyzed with a flow cytofluorimeter (Attune, Thermo Fisher Scientific, Waltham, MA, USA). The analyzed parameters were the percentage of cells expressing a given molecule and the mean fluorescence intensity (MFI). Dead cells and doublets were excluded from the analysis according to specific morphological parameters for size (forward scatter, FS) and granulation (side scatter, SS). The signal overflow between the channels for fluorescence detection was compensated before each experiment by using monochrome controls. Nonspecific fluorescence was determined by using isotype Abs and fluorescence minus one control (FMO). For the detection of intracellular molecules and cytokines, neutrophils were fixed and permeabilized with a fixation/permeabilization kit (Biolegend, San Diego, CA, USA) before staining with antibodies.

### 2.7. Determination of Inflammatory Cytokines and NE in Culture Supernatants

Supernatants from unstimulated neutrophil cultures, both groups of subjects, and stimulated cultures were used to determine the levels of cytokines and NE. The following stimuli were used to stimulate the cell response: OpZym (10 µg/mL), LPS (200 ng/mL) in combination with N-fMLP (1 µM), and inactivated opsonized *P. aeruginosa* (MOI 10). Cytokine concentrations were determined simultaneously by using fluorescent beads labeled with biotinylated anti-cytokine antibodies (LEGENDplex Human Inflammation Panel 1 kit, BioLegend, San Diego, CA, USA) on a flow cytofluorimeter (Attune, Thermo Fisher Scientific, Waltham, MA, USA) according to the manufacturer’s instructions. The cytokine concentrations were determined based on known cytokines in the standard of the kit. NE levels were measured in culture supernatants by using a commercial kit (R&D Systems, Minneapolis, MN, USA) according to the manufacturer’s instructions.

### 2.8. Assessment of Phagocytosis

Opsonized *P. aeruginosa* (MOI 10) labeled with the fluorescent dye carboxyfluorescein succinimidyl ester (CFSE) (2 µM) (Invitrogen/Thermo Fisher Scientific, Waltham, MA, USA) was added to neutrophil granulocytes (3 × 10^5^ per sample) that were previously stimulated with LPS (100 nM) and fMLP (1 µM). Live bacteria were preincubated at 70 °C (30 min) and then opsonized with 50% inactivated human serum for a 30-min incubation. The concentration of bacteria was determined on a spectrophotometer. After incubation (45 min at 37 °C), 0.2% Tripan solution was added to each sample to quench the fluorescence of extracellularly attached bacteria. The intensity of phagocytosis, expressed as % CSFE-positive cells, was determined with a cytofluorimeter (Attune, Thermo Fisher Scientific, Waltham, MA, USA).

### 2.9. Production of NETs

DNA staining with cell-impermeable dye (Sytox green (Invitrogen/Thermo Fisher Scientific, Waltham, MA, USA)) was used to follow the kinetics of the formation of NETs incubated with or without specific stimuli. Neutrophil granulocytes (1 × 10^5^/150 µL) in HBSS medium enriched with 1% human inactivated serum and 10 mM HEPES were seeded in a 96-well black flat-bottomed plate and incubated for 30 min at 37 °C. After incubation, cells were stimulated with various stimuli: PMA (50 nM), CaI (1 µM), OpZym (10 µM), fMLP (1 µM), and *P. aeruginosa* (MOI-10). TritonX-100 (1%) (Sigma-Aldrich, Steinheim, Germany) was added for neutrophil membrane permeabilization. After a four-hour incubation, a fluorescent dye, Sytox green, was added at a final concentration of 50 nM. The cells were incubated for 15 min at 37 °C. The fluorescence intensity, which was directly proportional to the degree of NETs, was read on a fluorimeter (Synergy HTX, Bio Tek Instruments, Santa Clara, CA, USA).

### 2.10. NET-Mediated Bacterial Killing 

The NET-mediated bacterial killing was performed as described [33] with a slight modification. Neutrophils (1 × 10^6^ per well) were placed in 24-well tissue culture plates and incubated with PMA (100 nM) for 4 h at 37 °C to stimulate NET release in the presence and absence of the phagocytosis inhibitor cytochalasin D (10 μg/mL; Sigma-Aldrich, Steinheim, Germany). After that, the supernatants were discarded, and deoxyribonuclease (DNase) was added to the corresponding wells. After 45 min, live *P. aeruginosa* (MOI-0.1) was placed in the wells and incubated for 1 h. After intensive mixing, 30 µL of supernatants with bacteria of different dilutions (1:10, 1:100, 1:1000) were taken from each well and inoculated on nutrient media (MacConkey agar). Bacterial colonies were counted after 48 h of incubation. The reduction in bacterial colonies in wells with NETs was calculated based on the number of bacterial colonies in the control wells. 

### 2.11. Statistics 

To describe the distribution of data, the mean and standard deviation for quantitative variables and absolute and relative frequencies for categorical variables were used. After testing the assumption of normality with the Kolmogorov–Smirnoff test with the Dallal–Wilkinson–Lilliefor *p*-value, group differences were tested with either the Student *t*-test for independent groups (normal distribution of variables) or the Mann–Whitney *U* test in the case of non-normality. In the case of multiple comparisons between the groups with normally distributed values, one-way ANOVA with the Bonferroni post-test was used. Associations between the quantitative variables were estimated by calculating Spearman’s rank correlation coefficients. *p*-values that were less than 0.05 were considered statistically significant. All analyses were performed with the statistical software SPSS 22.0 for Windows (released 2013; IBM Corp, Armonk, NY, USA).

## 3. Results

### 3.1. Clinical Characteristics of CF Patients

The clinical characteristics of CF patients are shown in detail in Table 2. The methodology for evaluating the clinical characteristics and complications of the disease is presented in Section 2.1. The patients had dominantly normal and mild severity of the disease in the lungs, with onset in the first year and good disease control. The mean forced expiratory volume in the first second (FEV1) and forced vital capacity (FVC) were similar (78.81 ± 24.40% and 78.89 ± 21.94%, respectively). Colonization with *P. aeruginosa* was more common than colonization with *Burkholderia cepacia*. Bronchopulmonary aspergillosis was the most common complication, whereas pulmonary hypertension was the rarest complication.

### 3.2. ROS Production

The production of ROS by neutrophils from CF patients stimulated with PMA or OpZym was statistically significantly lower in CF patients compared to that in the control neutrophils (*p* < 0.05). When other stimuli were used, such as the combination of fMLP and LPS, CaI, or *P. aeruginosa*, no significant differences were observed (Figure 1). Spontaneous ROS production by both CF and control neutrophils was very low. 

### 3.3. Apoptosis 

The spontaneous apoptosis of CF neutrophils was statistically significantly lower than that of the control neutrophils (Figure 2). All applied stimuli (fMLP/LPS, *P. aeruginosa*, or OpZym) significantly decreased the apoptosis of the control neutrophils compared to that of the non-stimulated cells (*p* < 0.05). However, adding *P. aeruginosa* to the culture of CF neutrophils decreased their apoptosis compared to that of non-stimulated CF neutrophils (*p* < 0.05). *P. aeruginosa* also decreased the apoptosis of CF neutrophils compared to that of control neutrophils that were treated similarly (*p* < 0.05). The differences in the apoptosis rates between CF and control neutrophils in the presence of other applied stimuli were not statistically significant (Figure 2).

### 3.4. NET-Osis 

NET-osis was increased in the CF and control neutrophils upon the stimulation of the cells in culture with PMA. Other treatments did not significantly modify NET-osis. However, there were no significant differences in spontaneous or stimulated (PMA, CaI, *P. aeruginosa*) NET-osis between the CF and control neutrophils (*p* > 0.05) (Figure 3A). When the CF patients were divided according to the severity of the disease, neutrophils from those with moderate–severe CF treated with *P. aeruginosa* showed a higher degree of NET-osis compared to that of normal (stable) and mild CF or control neutrophils that were stimulated with *P. aeruginosa* (Figure 3B). The differences were not statistically significant when other stimuli were used. In this context, there was a negative correlation (r = −0.5603; *p* < 0.05) between the NET-osis of *P. aeruginosa*-stimulated CF neutrophils and FEV1 (Figure 3C). This correlation was not seen with other stimuli of CF neutrophils.

### 3.5. Phagocytosis and Bactericidal Activity of NETs 

The phagocytosis of dead fluorescently labeled *P. aeruginosa* by fMLP/LPS-stimulated neutrophils was investigated via flow cytometry. The results presented in Figure 4A show no significant differences between CF and control neutrophils. However, there was a negative correlation between phagocytosis and the production of ROS by CF neutrophils in patients with CF stimulated with PMA (r = −0.499, *p* < 0.05), fMLP/LPS (r = −0.476, *p* < 0.05), and OpZym (r= −0.536, *p* < 0.05) (Figure 4B). A negative correlation existed in the healthy controls only when using OpZym as a stimulus of neutrophils (r = −0.648, *p* < 0.05) (Appendix A). 

There was a positive correlation between the intensity of NET-osis and phagocytosis of unstimulated CF neutrophils (r = 0.521, *p* < 0.05) and neutrophils stimulated with *P. aeruginosa* (r = 0.618, *p* < 0.05) but a negative correlation between these parameters in PMA-stimulated neutrophils (r = −0.540, *p* < 0.05) (Figure 4C). A positive correlation was also observed in unstimulated neutrophils of healthy controls (r = 0.536, *p* < 0.05) and healthy neutrophils stimulated with CaI (r = 0.533, *p* < 0.05) (Appendix A). No other correlations were statistically significant. The average value of the bactericidal activity of NETs obtained by PMA-stimulated neutrophils of children with CF against *P. aeruginosa* was 55.59%, and it did not significantly differ from the value in the control group (58.30%) (*p* > 0.05) (Figure 4D).

### 3.6. Secretion of NE by CF Neutrophils

The secretion of NE was significantly lower in non-stimulated neutrophils and fMLP-/LPS-stimulated neutrophils in CF patients compared to that in the healthy controls. There was no difference in NE production between the patients and healthy controls when other stimuli (PMA, OpZym, *P. aeruginosa*, and CaI) were used (Figure 5A).

There was a positive correlation between NE production and the intensity of phagocytosis of unstimulated CF neutrophils (r = 0.584, *p* < 0.05), as well as CF neutrophils stimulated with PMA (r = 0.828, *p* < 0.05), fMLP/LPS (r = 0.604, *p* < 0.05), OpZym (r = 0.793, *p* < 0.05), *P. aeruginosa* (r = 0.590, *p* < 0.05), and CaI (r = 0.734, *p* < 0.05) (Figure 5B). A positive correlation was also observed with control neutrophils (Appendix A). The correlation between the intensity of NET-osis and the NE levels in the CF neutrophil culture supernatants was not statistically significant, in contrast to the positive correlation observed in the control neutrophils (Appendix A).

### 3.7. Phenotypic Characteristics of CF Neutrophils

The expression of a large number of cell surface adhesion and chemokine molecules (CD11b, CD15, CD11c, CD44, CD54, CCR5), immunomodulatory molecules (PDL1, ILT3, IDO-1, ILT4), molecules involved in antigen recognition (Dectin1, CD206, TLR2, CD14), and molecules important for neutrophil function, costimulation, and antigen presentation (CD16, HLA-DR, MPO, FASL, CD86, NLRP3) on CF neutrophils was analyzed by using flow cytometry and compared with that of control neutrophils. As shown in Figure 6 and Appendix A, the expression of CD11b was higher on CF neutrophils stimulated with fMLP/LPS compared to the corresponding control neutrophils (*p* < 0.05), whereas the differences were not statistically significant for other markers.

### 3.8. Production of Cytokines and Chemokines by Neutrophils

The levels of pro-inflammatory cytokines and chemokines, including IL-8, IL-18, and MCP-1, were analyzed in the culture supernatants of neutrophils because other cytokines, such as IL-1β and Interferon-α (IFN-α), IL-6, TNF-α, IFN-γ, IL-33, and IL-23), were undetectable. Both non-stimulated (*p* < 0.005) and stimulated (fMLP/LPS, OpZym, *P. aeruginosa* ) (*p* < 0.01) neutrophils in children with CF produced significantly lesser amounts of IL-18 than those of the control neutrophils (Figure 7A). In contrast, unstimulated and *P. Aeruginosa*-stimulated CF neutrophils produced higher levels of IL-8 compared to the corresponding cultures of control neutrophils from healthy children (*p* < 0.05) (Figure 7B). There was a positive correlation (r = 0.498, *p* < 0.05) between the levels of IL-8 in the CF neutrophil culture stimulated with *P. aeruginosa* and FEV1 in CF patients (Figure 7C). The correlation was not observed with other stimuli. The levels of MCP-1 did not show statistically significant differences (Figure 7D).

A number of cytokines, including IL-1β, IL-4, IL-6, IL-10, IL-17A, TNF-α, and transforming growth factor (TGF)-β, were analyzed at the intracellular levels by using flow cytometry. As presented in Figure 8, only the expression of IL-10 was significantly higher in CF neutrophils than in the control neutrophils (*p* < 0.01), whereas the differences in other cytokines were not statistically significant (Appendix A).

## 4. Discussion

This is the first basic cross-sectional clinical and laboratory study of CF in children and adolescents from Bosnia and Herzegovina. The primary aim was to study the function and phenotype of neutrophils from peripheral blood. The reason is threefold: Neutrophils from BAL of CF are studied much more often; functional abnormalities of blood CF neutrophils are less pronounced than those of neutrophils isolated from BAL, and the results are very often inconsistent; no studies have examined abnormalities in peripheral blood neutrophils in children with CF systematically. Our results showed that the phenotypic characteristics of CF neutrophils are changed much less than their functions. Regarding functions, the most important changes were observed at the level of ROS production, followed by dysregulations of some pro- and anti-inflammatory cytokines and the production of NE. In contrast, phagocytosis, apoptosis, NET-osis, and anti-bacterial effects of NETs were normal or much less impaired. 

It is well known that CF patients have higher oxidative stress in the lungs than healthy subjects due to increased ROS production, which is simultaneous with a deficit of antioxidant molecules, and these abnormalities, which are partly associated with GSH deficiency due to low CFTR activity, play a fundamental role in the progression of chronic lung damage [34]. Similarly, elevated levels of oxidative stress markers were found in the serum of CF patients, and the oxidative stress progressively increased over the years and correlated with the severity of the disease [35]. ROS is also important for killing pathogens with neutrophils entrapped inside the phagolysosomal vacuole [35]. In this context, excessive activation of the neutrophil nicotinamide adenine dinucleotide phosphate oxidase (NOX2) results in exaggerated ROS release in the airway microenvironment, which increases oxidative damage to lung tissues [36]. A reduction in ROS production and functional exhaustion of CF airway neutrophils was also described [9]. This relation to CF blood neutrophils is controversial, as these cells showed both higher and normal ROS production [37,38,39] or this process depended on infecting pathogens [40,41]. Our study found a significantly lower intensity of oxidative burst of blood neutrophils in CF patients than in the control group when stimulating cells with PMA and OpZym. These results may correspond to the findings obtained for activated neutrophils occurring in the lung microenvironment of CF patients. The non-modified ROS production in the presence of *P. aeruginosa* was in line with the finding that this bacterial species did not significantly modify oxidative stress [35].

The ability of neutrophils, as short-lived cells, to rapidly die under physiological conditions by apoptosis is protective for the host and is a key mechanism for the resolution of inflammation [42]. However, extravasated neutrophils have a prolonged life span during inflammation due to inhibited apoptosis. Similar findings were detected in the lungs of CF patients [43]. In contrast, some studies showed that the number of apoptotic neutrophils was increased in the sputum of CF patients, especially in the presence of *P. aeruginosa* [44]. The findings in peripheral blood seem different because CF neutrophils have decreased apoptosis [45,46,47,48]. In another study, neutrophils isolated from CF patients showed enhanced survival and upregulation of p21/Waf1 (a cyclin-dependent kinase inhibitor), and increased expression of proliferating cell nuclear antigen (PCNA) [49]. Delayed apoptosis in CF neutrophils could be due to systemic inflammation [45], loss of some CFTR functions, or interaction of CFTR with other molecules regulating neutrophil functions [20,46,47,48]. Our study demonstrated a significantly lower degree of apoptosis of CF neutrophils (non-stimulated and stimulated with *P. aeruginosa*). The findings on non-stimulated CF neutrophils support the hypothesis that delayed apoptosis is associated with the CFTR defect. However, increased survival of these cells in the presence of *P. aeruginosa* is not in accordance with the findings that some products of this bacteria, such as pyocyanin, induce neutrophil apoptosis due to rapid and prolonged ROS generation and subsequent reduction of intracellular cAMP [50]. Therefore, decreased apoptosis of CF neutrophils may be indirect and caused by pro-inflammatory mediators such as IL-8 triggered by *P. aeruginosa* in neutrophils, as we showed in this study. This chemokine is known to be a survival factor for neutrophils [51]. 

NET-osis is one of the additional mechanisms of the anti-bacterial activity of neutrophils. Extruded NETs comprise DNA, histones, enzymes (NE and MPO), and antibacterial and proinflammatory molecules [52]. NET-osis is important for the pathophysiology of lung disease in CF, and in this context, an excess of NETs has been described in airways with CF [53,54,55]. It was proposed that NET formation can promote CF inflammation through interactions with macrophages, which become overstimulated and produce pro-inflammatory mediators [43]. Gray et al., 2018 [46] demonstrated an enhanced NET formation of CF blood neutrophils in culture upon the cells’ prolonged cultivation (6 h), and they were then induced to form NETs. This finding was not observed after a shorter period of incubation (4 h), as in our study. *P. aeruginosa* is a significant modulator of NET-osis, and some factors from this bacteria can enhance or suppress NET-osis. The reduction of NET-osis by *P. aeruginosa* resulted from the sialic acid binding from the bacteria’s surface to neutrophils [56]. In contrast, pyocyanin, a virulence factor of *P. aeruginosa*, enhances NET formation and requires NOX2 for its action [57]. In this context, NET-mediated resistance to killing *P. aeruginosa* has evolved over time, and this phenomenon was correlated with the development of the mucoid phenotype [58,59]. We showed a higher degree of NET-osis triggered by *P. aeruginosa* in a small group of children with a moderate–severe form of CF, suggesting that NET-osis could be a significant mechanism of disease exacerbation. 

NE is a component of NET. Unexpectedly, we found significantly lower levels of NE in non-stimulated and fMLP-/LPS-stimulated neutrophil cultures in children with CF. Up to now, no such studies have been published. However, we did not find a correlation between the NE levels and the NET-osis intensity. NE is a serine protease stored in azurophilic granules, which are released during degranulation, NET formation, or cell death [60]. The enzyme is a major product released from neutrophils in inflamed airways in CF and represents a key risk factor for the onset and early progression of CF lung disease [61]. Previous findings suggested that clinically stable adult CF patients colonized with *P. aeruginosa* had higher NE levels in both plasma and sputum compared to those of CF patients without *P. aeruginosa* infection. One study showed that free NE in sputum correlated with FEV1 in children with CF [62]. Another study reported that FEV1 inversely correlated with total cell count and free NE in BAL fluid obtained from patients with CF [63]. The lower production of NE by CF neutrophils in our study could be due to the stronger binding of the enzyme to NETs [64]. Alternatively, NE activity could be blocked by some inhibitors, as shown in some experiments in CF lung disease [65], or neutrophil proteases could degrade NE. Phagocytosis is a crucial mechanism of the anti-bacterial activity of neutrophils and macrophages. Dysfunction of phagocytosis in CF could be a reason for bacterial colonization, which is partly due to the reduced expression of cell surface recognition receptors [66]. This finding is also accompanied by defective intraphagolysosomal bacterial killing due to the absence or dysfunction of CFTR at the level of phagolysosomes [17]. Morris et al. (2005) demonstrated that neutrophils from patients with CF had a lower phagocytic capacity than that of circulating neutrophils from the same patients or from normal control subjects [67]. Mortaz et al. (2019) showed that neutrophils from peripheral blood in children with CF did not show impaired phagocytosis of *Staphylococcus aureus* and *P. aeruginosa* [68], as we showed in our study. We did not find differences in NET-mediated killing of *P. aeruginosa* between CF and control neutrophils, in contrast to some other results obtained with a different methodology [69]. The findings in sputum seem to be different because an increased pathogen resistance in response to NETs was reported [64]. 

In our study, we analyzed the expression of 20 cell surface and intracellular molecules on CF and control neutrophils, including adhesion molecules, chemokine receptors, immunomodulatory molecules, receptors involved in antigen recognition, and molecules that are important for neutrophil function, costimulation, and antigen presentation. This is one of the largest panels of molecules analyzed in CF neutrophils. However, the only significant finding was a higher expression of CD11b compared to that in control neutrophils. Until now, upregulation of CD11b, CD66b, CD63, HLA-DR, and CD80 and down-regulation of CD16 and CD14 in neutrophils in the BAL of CF patients in comparison with blood CF neutrophils was observed [70]. Both CD14 and CD16 are essential for phagocytosis, and their loss, along with that of CXCR1 [71], could be partly responsible for the inability of CF airway neutrophils to kill bacteria [72]. In another study, airway neutrophils in CF patients showed an increase in the expression of CD63 (a marker of elastase exocytosis) and decreased CD16 due to its shedding from neutrophils [73]. It is well known that neutrophil migration from the bloodstream involves alterations in the surface expression of the adhesion molecules (CD62L and CD11b/CD18) [74]. Russell et al. (1998) showed that their basal levels in blood neutrophils were similar both in CF and healthy subjects. However, upon activation with IL-8 or fMLP, upregulation of CD11b was observed [75], which was similar to what we obtained in the presence of fMLP/LPS. Recent data [76] have shown a significant increase in the CD11b expression in neutrophils from CF patients during exacerbation of the disease due to an increase in the percentage of a neutrophil subset named low-density neutrophils (LDNs), which are characterized by the CD16^high^/CD62L^low^ phenotype. 

We examined the production of cytokines—predominantly those with pro-inflammatory properties—in the supernatants of neutrophil cultures, such as IL-8, MCP, IL-18, IL-1β, IFN-α, IL-6, TNF-α, IFN-γ, IL-33, and IL-23. Of all cytokines, only IL-8, IL-18, and MCP-1 were measurable, independently of whether the cells were stimulated or not. Anti-inflammatory cytokines (TGF-β, IL-4, and IL-10) and pro-inflammatory cytokines (IL-1β, IL-6, TNF-α, IL-17, and IL-33) were measured intracellularly. These cytokines are involved in neutrophil recruitment, activation, and NET-osis [70]. Our results showed that the production of IL-8 by unstimulated and *P. aeruginosa*-stimulated neutrophils was higher in children with CF than in the controls. IL-8 plays a central role in the pathophysiology of CF disease since its excessive production and retention in BAL are followed by robust infiltration of neutrophils in the lungs, even in the absence of lung disease exacerbation [77,78]. Increased levels of IL-8 were detected both in the BAL and serum of CF patients [1,38,70]. Montemurro et al. (2012) showed that circulating neutrophils of pediatric and adult CF patients with acute exacerbation of lung disease produced greater amounts of IL-8 than in healthy controls and, unexpectedly, the levels further increased upon antibiotic therapy [38]. Another study also demonstrated that neutrophils from pediatric CF patients showed increased IL-8 production in acute exacerbation, which was not further potentiated by LPS [19]. The association of spontaneously increased IL-8 production and GFTR deficiency was confirmed by the decreased production of this chemokine upon therapy with CFTR-modulating drugs [79].

The results related to the production of IL-18 by CF neutrophils, which are opposite to those obtained for IL-8, are very interesting and have not been published until now. However, the results published for other cell models are contradictory. For example, increased expression of IL-18 in macrophages and epithelial cells in the lungs, with simultaneously decreased levels in the CF sputum, was found [80], suggesting that certain factors in BAL, such as exotoxin from *P. aeruginosa*, can degrade this cytokine [81]. Scambler et al. (2019) identified increased levels of IL-18, IL-1β, and caspase-1 activity in the monocytes, epithelia, and serum of CF patients, and these were reversed by pretreatment with NLRP3 inflammasome inhibitors [82]. Other opposite results for BAL cells, PBMC, and serum levels of IL-18 were published [83]. These results were correlated with increased IL-10 production and decreased Th1 response because IL-18 is among the key IFN-γ-stimulating factors [83]. Our results support this hypothesis because upregulation of both IL-8 and IL-10 could down-modulate the expression of IL-18 [80]. The decreased serum levels of IL-18 in CF could be due to the up-regulation of the IL-18-binding protein [84], a decoy receptor for IL-18 [85]. Alternatively, it is known that IL-18 is prone to degradation with NE in BAL fluid in CF when it is not bound to glycosaminoglycans (GAGs) or other extracellular matrix components. With higher levels of IL-8, its competition for binding to GAGs is higher than that of IL-18 [86]. This mechanism can be operative in our culture supernatants of neutrophils in which both IL-8, NE and many extracellular components are present.

Our study has some limitations. First, the number of children with CF was too small for them to be divided into clinically different groups, as the number of participants per group could not be sufficient for a proper conclusion. This is the main weakness of this study. A comparative analysis of neutrophils from peripheral blood and BAL was impossible due to ethical principles because bronchoscopy is not indicated in patients with normal lung function. This problem was overcome by adding bacterial or pharmacological stimuli that mimicked the activation of neutrophils at the site of infection. Many stimuli have a dose-dependent effect, and due to the limited number of cells from blood sampling, there was no possibility for such testing. However, the results obtained by using complex functional assays and multiple phenotypic markers (the main strength of the study) are a good starting point for extending research in the next experiments in at least two directions: a comparison of neutrophil dysfunction with CF severity in a larger cohort of children and the monitoring of neutrophil functions after therapy with CFTR modulators (a study that is in progress). 

## 5. Conclusions

Peripheral blood neutrophils from children with CF showed fewer impaired changes in phenotype than in function. The functional abnormalities of neutrophils detected at the baseline levels were additionally potentiated or induced by adding stimuli in culture. In comparison with healthy children, *P. aeruginosa* inhibited apoptosis and IL-18 but stimulated IL-8 production (whole group) and NET-osis in the moderate–severe subgroup of CF patients. fMLP/LPS inhibited NE and IL-18 production and up-regulated the expression of CD11b and IL-10. PMA and OpZym inhibited ROS production, and OpZym additionally inhibited IL-18. In non-stimulated CF neutrophils, NE and IL-18 production was inhibited, whereas IL-8 was up-regulated. Cumulatively, these results (presented additionally in the form of a graphical abstract) suggest that neutrophil dysfunction in children with CF is already present in peripheral blood. However, these changes are more visible after the addition of stimuli that differently mimic the activation of these cells at the site of infection. The obtained results can help monitor the disease severity or the effectiveness of the therapy.

## Figures and Tables

**Figure 1 biomedicines-11-01725-f001:**
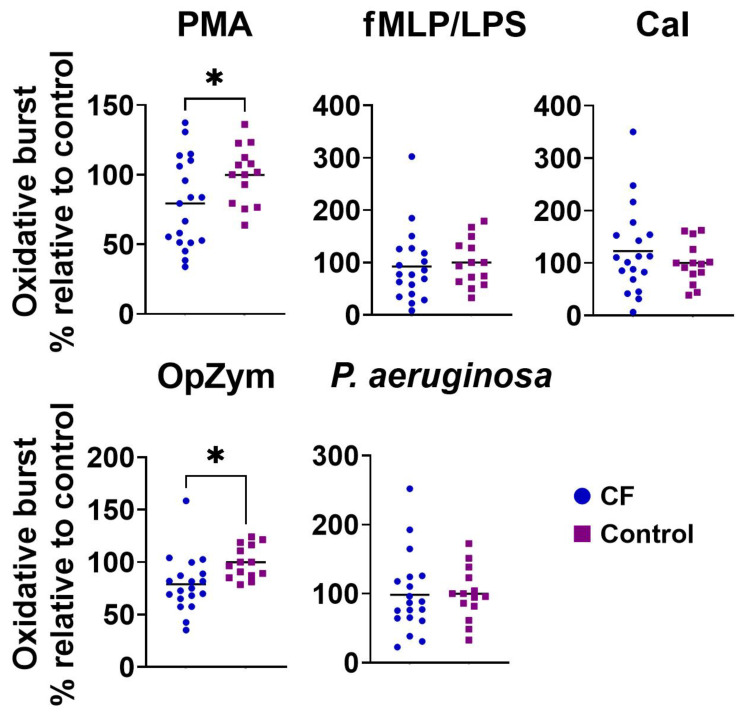
The oxidative burst of peripheral blood neutrophils from children with cystic fibrosis (CF) and control children. Oxidative burst was performed as described in Section 2.4. Values are given as values relative to control (the values of oxidative burst for an individual patient or control child were divided by the mean control for that experiment). PMA—phorbol myristate acetate; fMLP/LPS—*N*-formyl-methionyl-leucyl-phenylalanine and lipopolysaccharide; CaI—calcium ionophore, OpZym—opsonized zymosan. * *p* < 0.05 compared to the control group; Student’s *t*-test.

**Figure 2 biomedicines-11-01725-f002:**
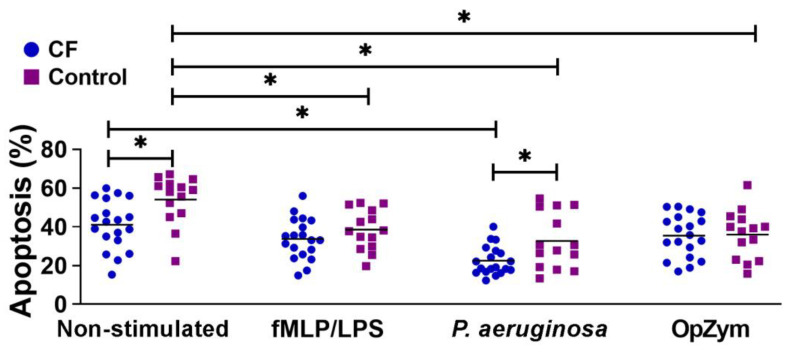
Apoptosis of peripheral blood neutrophils from children with cystic fibrosis (CF) and control children. Apoptosis was performed as described in Section 2.5. and is presented in percentages (*n* = 19 for CF; *n* = 14 for the control group). fMLP/LPS—*N*-formyl-methionyl-leucyl-phenylalanine and lipopolysaccharide; OpZym—opsonized zymosan. * *p* < 0.05 compared to the corresponding groups as indicated on the bars; Student’s *t*-test.

**Figure 3 biomedicines-11-01725-f003:**
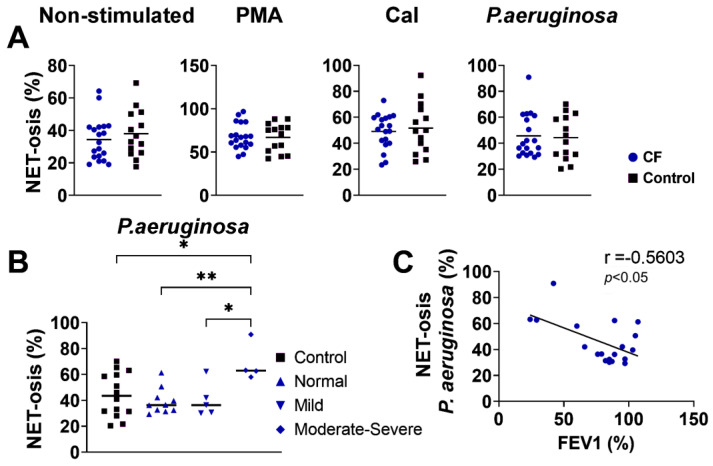
(**A**) NET-osis of peripheral blood neutrophils from children with cystic fibrosis (CF) and control children. NET-osis was performed as described in Section 2.9. and is presented in percentages (*n* = 19 for CF; *n* = 14 for the control group). (**B**). Comparison of NET-osis between CF patients divided according to the severity of the disease. (**C**). Correlation between NET-osis and forced expiratory volume in the first second (FEV1). PMA-phorbol myristate acetate; fMLP/LPS-*N*-formyl-methionyl-leucyl-phenylalanine and lipopolysaccharide; CaI-calcium ionophore, OpZym-opsonized zymosan. * *p* < 0.05; ** *p* < 0.01 compared to the corresponding groups as indicated; Student’s *t*-test (**A**), one-way ANOVA with Bonferroni post-test (**B**), and Spearman’s rank correlation test (**C**).

**Figure 4 biomedicines-11-01725-f004:**
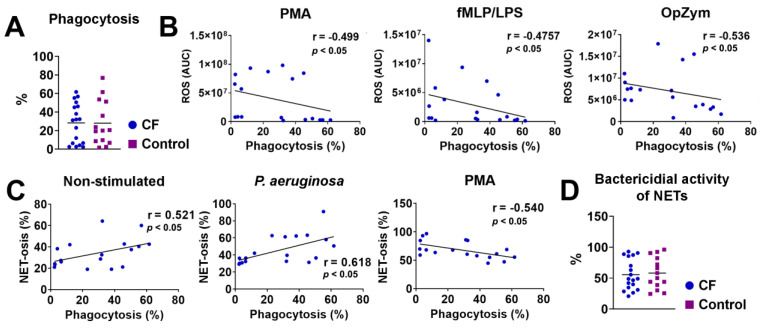
(**A**) Phagocytosis of *P. aeruginosa* by cystic fibrosis (CF) and control neutrophils. (**B**) Correlation between the ROS activity and phagocytosis of CF neutrophils. (**C**) Correlation between NET-osis and phagocytosis of CF neutrophils. (**D**) Bactericidal activity of NETs released from CF and control neutrophils. The detailed methodology is described in Section 2.10. PMA-phorbol myristate acetate; fMLP/LPS-*N*-Formyl-methionyl-leucyl-phenylalanine and lipopolysaccharide; OpZym-opsonized zymosan; Spearman’s rank correlation test (**A**–**C**) and Student’s *t*-test (**D**).

**Figure 5 biomedicines-11-01725-f005:**
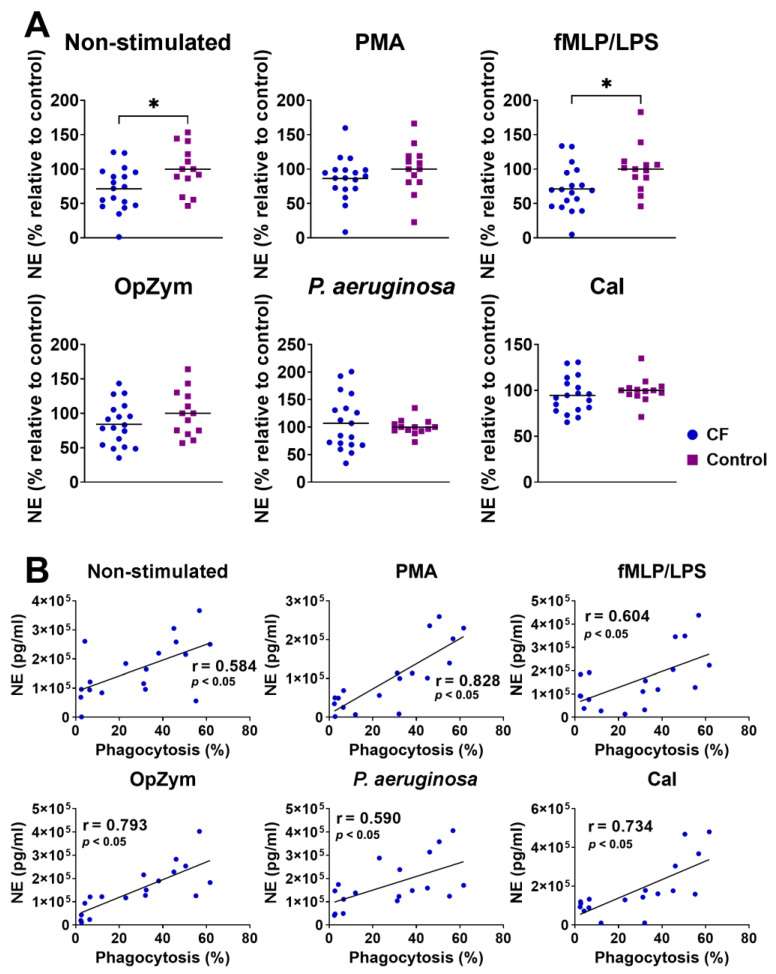
(**A**) Comparison of neutrophil elastase (NE) levels in non-stimulated and stimulated neutrophil cultures from children with cystic fibrosis (CF) and healthy children. NE detection was performed as described in Section 2.7. Values are given as values relative to the control (the values of NE for an individual patient or control child were divided by the mean control for that experiment). (**B**) Correlation of NE levels in supernatants of CF neutrophil cultures and phagocytosis. PMA—phorbol myristate acetate; fMLP/LPS-*N*-formyl-methionyl-leucyl-phenylalanine and lipopolysaccharide; CaI-calcium ionophore, OpZym-opsonized zymosan. * *p* < 0.05, compared to the corresponding controls; Student’s *t*-test (**A**) and Spearman’s rank correlation test (**B**).

**Figure 6 biomedicines-11-01725-f006:**
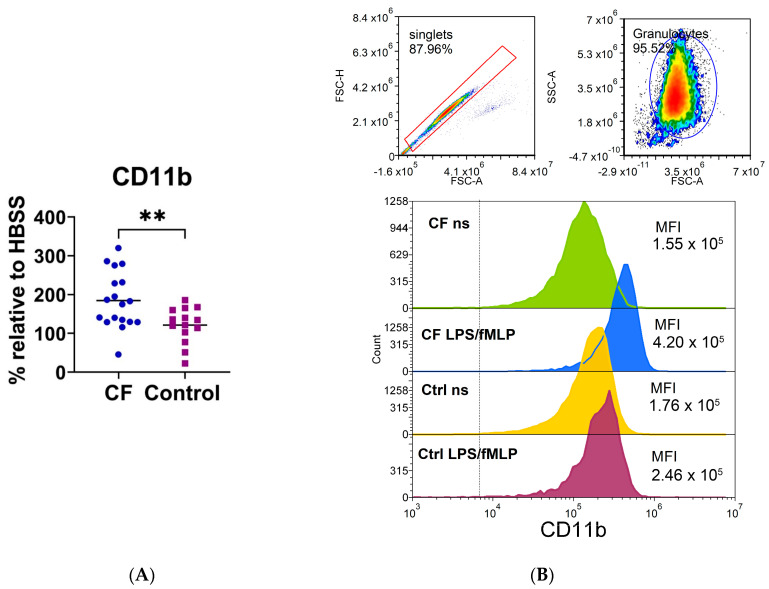
(**A**) The expression of CD11b on neutrophils stimulated with lipopolysaccharide and *N*-formyl-methionyl-leucyl-phenylalanine (LPS/fMLP). (**B**) A representative histogram of CD11b fluorescence. Upper part: doublet exclusion and gating strategy. Lower part: the fluorescence profile of cystic fibrosis (CF) and control (Ctrl) neutrophils; non-stimulated (ns) or LPS-/fMLP-stimulated cells. MFI—mean fluorescence intensity; ** *p* < 0.01, compared to the control. Results are presented as the percentage change in mean MFI of stimulated neutrophils compared to non-stimulated cells (basal fluorescence of neutrophils in HBSS medium); Student’s *t*-test.

**Figure 7 biomedicines-11-01725-f007:**
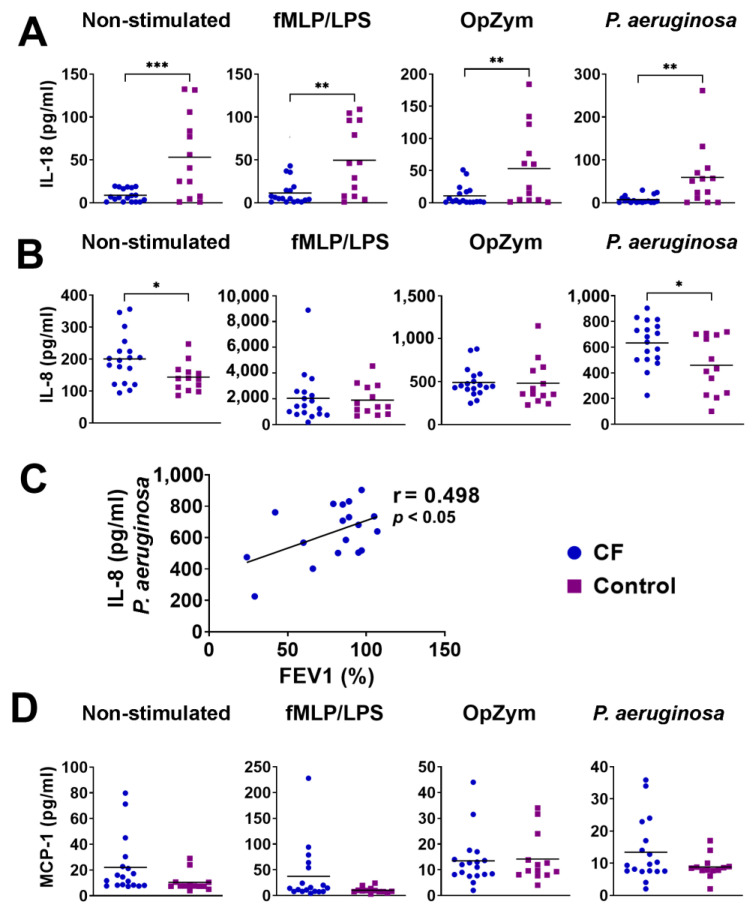
The levels of IL-18 (**A**) and IL-8 (**B**) in the culture supernatants of neutrophils. (**C**) Correlation between IL-8 in culture supernatants and forced expiratory volume in the first second (FEV1). (**D**) The levels of monocyte chemoattractant protein-1 (MCP-1) in culture supernatants. * *p* < 0.05; ** *p* < 0.01; *** *p* < 0.005, compared to the corresponding controls. fMLP/LPS-*N*-formyl-methionyl-leucyl-phenylalanine and lipopolysaccharide; OpZym-opsonized zymosan; Mann–Whitney *U* test (**A**,**B**,**D**) and Spearman’s rank correlation test (**C**).

**Figure 8 biomedicines-11-01725-f008:**
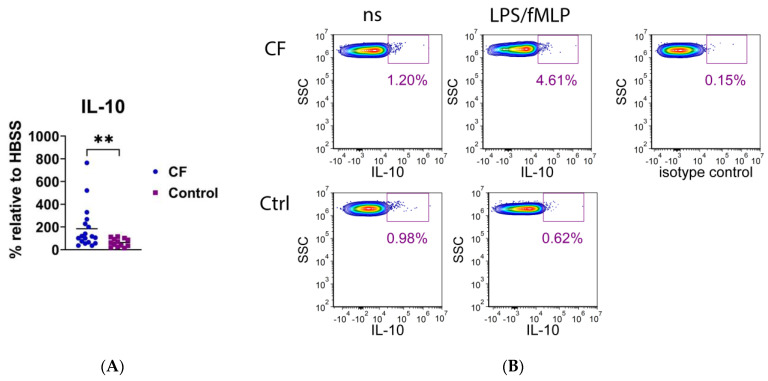
(**A**) Intracellular expression of IL-10 in neutrophils stimulated with lipopolysaccharide and *N*-formyl-methionyl-leucyl-phenylalanine (LPS/fMLP). Results are presented as the percentage change in mean fluorescence intensity (MFI) of stimulated neutrophils compared to non-stimulated (ns) cells (basal fluorescence of neutrophils in HBSS medium). (**B**) A representative histogram of IL-10 fluorescence of CF and control (Ctrl) neutrophils. ** *p* < 0.01, compared to the control; Mann–Whitney *U* test.

**Table 1 biomedicines-11-01725-t001:** Demographic characteristics of the study participants.

Variables	Mean Years	SD	Minimum	Maximum
CF				
Total (*n* = 19)	12.53	3.89	5	17
Male (*n* = 12)	11.45	4.24	5	17
Female (*n* = 7)	13.43	3.55	8	17
BMI (kg/m^2^)	16.50	2.41	13.50	22.00
Control				
Total (*n* = 14)	13.90	3.11	6	16
Male (*n* = 8)	12.42	3.56	7	16
Female (*n* = 6)	14.80	2.96	8	15
BMI (kg/m^2^)	17.42	2.22	14.32	23.48

CF—cystic fibrosis; BMI—body mass index.

**Table 2 biomedicines-11-01725-t002:** The clinical characteristics of CF patients.

Variables			Total (*n* = 19)	
			*n*	%
The onset of disease symptoms				
0–30 days			6	31.6
30–365 days			8	42.1
>365 days			5	26.3
Initial clinical manifestations				
Meconium ileus			2	10.5
Non-progression syndrome			10	52.6
Recurrent broncho-obstructions			7	36.8
Age of diagnosis				
0–12 months			10	52.6
1–3 years			3	15.8
>3 years			6	31.6
Molecular genetics				
Homozygous			16	84.2
Compound heterozygous			3	15.8
Complications of bronchopulmonary aspergillosis				
No			11	57.9
Yes			8	42.1
Complications of pulmonary hypertension				
No			18	94.7
Yes			1	5.3
Complications of type I diabetes				
No			13	68.4
Yes			6	31.6
Chronic colonization with *Burkholderia cepacia complex*				
No			17	89.5
Yes			2	10.5
Chronic colonization with *P. aeruginosa*				
No			3	15.8
Yes			10	52.6
Intermittent			5	31.6
The severity of the lung disease				
Normal			10	52.6
Mild			5	26.3
Moderate and severe			4	21.1
Control of the lung disease				
Good control			14	73.7
Bad control			5	26.3
**Variables**	**Mean**	**SD**	**Minimum**	**Maximum**
FEV1 (%)	78.81	24.40	24.00	107.00
FVC (%)	78.89	21.94	32.00	124.00
FEV1—forced expiratory volume in the first second; FVC—forced vital capacity				

## Data Availability

All data are available on request from the corresponding author.

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
