# Peer review of "Dysfunctions of Neutrophils in the Peripheral Blood of Children with Cystic Fibrosis"

_biomedicines, 2023, doi:10.3390/biomedicines11061725_

Round 1

Reviewer 1 Report

Cystic fibrosis is a rare and severe genetic disease, and dysfunction of the immune system, in particular neutrophils, plays an important role. Infectious complications are common and severe, so knowledge of neutrophil function in cystic fibrosis will help select the relevant therapeutic approach. Though there are many articles devoted to the study of neutrophils in cystic fibrosis, the authors have performed complex investigation of the biochemistry and physiology of neutrophils in different aspects, by different methods and using several stimuli. All these makes it possible to create a holistic view of the problem. However, while reading the article, I had several questions. I hope that the answers to them will improve the presentation of scientific material.

Abstract

1) Please decipher FEV1 in abstract

Materials and methods

2) Please name subsection 2.1

3) In the control group, the authors say that the inclusion criterion was the absence of acute infectious diseases within a month and no signs of chronic diseases. Please specify if you mean allergies? Also check if the children have taken antioxidant supplements, which can also affect basal neutrophil activity.

4) The authors use a combination of fMLP and LPS stimuli. Describe, please, is this a sequential or simultaneous introduction of the stimuli? What is the rationale for using two stimuli instead of one? Please, give a relevant reference or describe in more detail this procedure.

Results

5) In Table 1, the minimum and maximum ages are indicated with an accuracy of several days (5.00, 17.00 etc). Is it really true? If the age is given with an accuracy of one year, then the number of significant digits should be 5, 17, 6, and 16.

6) Figure 1 (subsection 3.1). You write that the activity of zymosan-stimulated neutrophils in patients with cystic fibrosis is less than in the control group, and I agree. However, regarding the PMA, I have my doubts. It seems to me that in the PMA group, two subgroups can be distinguished - in half of the patients with cystic fibrosis, activity decreased, in the other it did not change. What do you think about it?

7) Fig. 1. The authors present data as percentages relative to the mean control. In my opinion, this is possible when the distribution of the parameter in the control group is normal. Please clarify this point.

8) Fig. 1. Specify which statistical test was used. Is it t-test or Mann-Whitney test? The same, please, for Figs. 2, 3, 5, 6, and 7.

Discussion

9) Remove the first phrases left from the template (Authors should discuss the results and how they can be interpreted from the perspective of previous studies and of the working hypotheses. The findings and their implications should be discussed in the broadest context possible. Future research directions may also be highlighted).

Conclusions

10) Please check all the conclusions. You write PMA and OpZym stimulated ROS production. This is true? Judging by Figure 1, they inhibit the respiratory burst in patients. “Pseudomonas aeruginosa inhibited apoptosis”, but I would clarify that they inhibit apoptosis in patients to a greater extent than in healthy ones. “Pseudomonas aeruginosa stimulated NETosis”, but I would clarify that in the group of severely ill patients. fMLP/LPS inhibited NE - is that so? Without the stimuli, NE is also reduced in the cystic fibrosis group.

11) Also in the conclusion, please, say several words about the practical medical significance of your study. Your findings may be used for disease monitoring, for example, or for monitoring the effectiveness of therapy.

Throughout the text:

12) Please correct typos (ninethin, line 315), check subscripts and superscripts (lines 193, 199, 211, 213, 223, 227, 229, 283, 284 etc), spaces, italics, etc.

Author Response

Reviewer 1:

Cystic fibrosis is a rare and severe genetic disease, and dysfunction of the immune system, in particular neutrophils, plays an important role. Infectious complications are common and severe, so knowledge of neutrophil function in cystic fibrosis will help select the relevant therapeutic approach. Though there are many articles devoted to the study of neutrophils in cystic fibrosis, the authors have performed complex investigation of the biochemistry and physiology of neutrophils in different aspects, by different methods and using several stimuli. All these makes it possible to create a holistic view of the problem. However, while reading the article, I had several questions. I hope that the answers to them will improve the presentation of scientific material.

General: The authors thank you very much for the valuable comments. The text is revised according to your suggestions and also the comments of two other reviewers. You can find now point-by-point answers to all your comments. Other modifications of the text (shortness of discussion, changing order of some results, moving of previous Table 3 to supplementary materials, better explanation and clarifications of some judgments are related to the comments of other reviewers.

Abstract

1) Please decipher FEV1 in abstract

Corrected

 Materials and methods

2) Please name subsection 2.1

Corrected, sorry for the mistake that occurred during the uploading of the sections

3) In the control group, the authors say that the inclusion criterion was the absence of acute infectious diseases within a month and no signs of chronic diseases. Please specify if you mean allergies? Also check if the children have taken antioxidant supplements, which can also affect basal neutrophil activity.

This explanation was added: The control group of children had normal physical findings, no signs of current acute infection nor infection during the last month, and no history of chronic diseases, including allergies. The participants from both groups taking oral steroids, antibiotics, or antioxidant supplements in the last month were excluded.

4) The authors use a combination of fMLP and LPS stimuli. Describe, please, is this a sequential or simultaneous introduction of the stimuli? What is the rationale for using two stimuli instead of one? Please, give a relevant reference or describe in more detail this procedure.

We explained the reason for this approach and cited a paper: The simultaneous addition of LPS, as a neutrophile priming agent, and fMLP, as a triggering event, is a standard procedure to induce ROS production or better stimulation of neutrophils  [32]. (lines 238-240)

 Results

5) In Table 1, the minimum and maximum ages are indicated with an accuracy of several days (5.00, 17.00 etc). Is it really true? If the age is given with an accuracy of one year, then the number of significant digits should be 5, 17, 6, and 16.

Corrected, sorry for this mistake

6) Figure 1 (subsection 3.1). You write that the activity of zymosan-stimulated neutrophils in patients with cystic fibrosis is less than in the control group, and I agree. However, regarding the PMA, I have my doubts. It seems to me that in the PMA group, two subgroups can be distinguished - in half of the patients with cystic fibrosis, activity decreased, in the other it did not change. What do you think about it?

As you can see the response (ROS production) differs between individuals both in CF and control groups and bars show mean values. Such differences are seen also in the presence of other stimuli, but statistical analysis confirmed that the differences existed only in OpsZym- and PMA-treated cultures. Interindividual variability in the response of neutrophils in many assays presented in this work is something that has already been published many times.

7) Fig. 1. The authors present data as percentages relative to the mean control. In my opinion, this is possible when the distribution of the parameter in the control group is normal. Please clarify this point.

Yes, the distribution of the parameter in the control group is normal. The differences in this assay were tested by using the Student t-test (this is added in Fig.1 legend).

8) Fig. 1. Specify which statistical test was used. Is it t-test or Mann-Whitney test? The same, please, for Figs. 2, 3, 5, 6, and 7.

The explanation is now given in each Fig. legend.

 Discussion

9) Remove the first phrases left from the template (Authors should discuss the results and how they can be interpreted from the perspective of previous studies and of the working hypotheses. The findings and their implications should be discussed in the broadest context possible. Future research directions may also be highlighted).

Corrected, sorry for the mistake that occurred during the uploading of the sections

Conclusions

10) Please check all the conclusions. You write PMA and OpZym stimulated ROS production. This is true? Judging by Figure 1, they inhibit the respiratory burst in patients. “Pseudomonas aeruginosa inhibited apoptosis”, but I would clarify that they inhibit apoptosis in patients to a greater extent than in healthy ones. “Pseudomonas aeruginosa stimulated NETosis”, but I would clarify that in the group of severely ill patients. fMLP/LPS inhibited NE - is that so? Without the stimuli, NE is also reduced in the cystic fibrosis group.

Our intention was to make the conclusion relevant only to the CF group. So we now clarified this conclusion, corrected the obvious mistake (thank you very much), and added a sentence for non-stimulated CF neutrophils. The conclusion is now as follows:

Peripheral blood neutrophils from children with CF showed less impaired changes in phenotype than function. Functional abnormalities of neutrophils detected at the baseline levels were additionally potentiated or induced by adding stimuli in culture. In comparison with healthy children, P. aeruginosa inhibited apoptosis and IL-18 but stimulated IL-8 production (whole group) and NET-osis in the moderate-severe subgroup of CF patients. fMLP/LPS inhibited NE and IL-18 production and up-regulated the expression of CD11b and IL-10. PMA and OpZym inhibited ROS production, and OpZym additionally inhibited IL-18. In non-stimulated CF neutrophils NE and IL-18 production was inhibited, whereas IL-8 was up-regulated. Cumulatively, these results (presented additionally in the form of a graphical abstract) suggest that neutrophil dysfunction in children with CF is already present in peripheral blood. However, these changes are more visible after the addition of stimuli that differently mimic the activation of these cells at the site of infection. The obtained results can help to monitor the disease severity or the effectiveness of the therapy.

11) Also in the conclusion, please, say several words about the practical medical significance of your study. Your findings may be used for disease monitoring, for example, or for monitoring the effectiveness of therapy.

The conclusion was finished with the following sentence: The obtained results can help to monitor the disease severity or the effectiveness of therapy.

Throughout the text:

12) Please correct typos (ninethin, line 315), check subscripts and superscripts (lines 193, 199, 211, 213, 223, 227, 229, 283, 284 etc), spaces, italics, etc.

Thank you, We have checked the whole text and corrected the obvious mistakes.

Reviewer 2 Report

The manuscript “Dysfunctions of neutrophils in peripheral blood in children with cystic fibrosis” describes a prospective clinical study performed in 19 children with cystic fibrosis and 14 age-matched controls. The aim of the study was to evaluate the functional differences in peripheral blood neutrophils in children with cystic fibrosis compared to children without cystic fibrosis. The authors measured a large number of phenomena and stimulated the isolated neutrophils with several different inducers. This is of course of general interest to the field and an interesting alternative approach to understand what happens in cystic fibrosis. The experiments are carefully described and the text well written, with some minor errors in language.

There are some concerns with the manuscript which need to be addressed:

Major issues:

1: Figure 3: How can four values be enough to give statistical significance compared to three other groups? If correcting for multiple comparisons, there is no way this could be the case. Please use the correct statistical methods. On this same note, please indicate in all figure legends what statistical methods were used.

2: Page 4, line 152: Classification of lung damage: it would strengthen the grouping to include high resolution chest CT for evaluation of lung damage.

3: The discussion is too long. There is no way that the experiments performed can have such great impact that they merit such a long discussion. Please keep it short and to the point. It would also be of value if you briefly discussed the strengths and weaknesses of the study.

4: Why did you not grow the Pseudomonas aeruginosa under mucoid conditions?

Minor issues:

1: The language needs some improvement and clarification:

Abstract, line 21: lesser examined. Please change to “less examined”.

Page 2, line 90: Dysphynction. Please change to ”dysfunction”, if that is what you mean.

Page 8, line 315: What does “Ninethin” mean? Nineteen?

2: Please explain some statements made:

Page 3, line 110: Do you mean to say “2.1. Subsection”? The heading should read ”selection of participants” or something similar.

Page 4, line 159: What do you mean by “and its control”?

Page 5, line 216: What is calcium ionophore (CaI)? It must have a name.

Page 5, line 242: What does “cultured as indicated for apoptosis” mean? Please describe how this was done.

Page 17, line 521: Apoptosis is the key biological function in the host’s immune response and inflammation. Please remove this statement as it is confusing.

Page 17, line 542: patients with gating (G551D) mutations. Either the patients have G551D or some other gating mutation (or both, on separate alleles). What do you mean?

3: Please correct the numbering of the tables:

Table 3 should be table 1.

Table 2 should be in section 2.2 and be called table 1.

Please make sure you have numbered the tables correctly and the text refers to the correct table. The demographics are not results but should be presented in the materials-section.

4: Please correct the text:

The acronym should be introduced the first time the substance or expression is mentioned, for example LPS. Then the acronym should be used and not explained again.

Please remove the first sentences from the discussion (Authors should discuss the results and how they can be interpreted from the perspective of previous studies and of the working hypotheses. The findings and their implications should be discussed in the broadest context possible. Future research directions may also be highlighted.) This is not an instruction for how to write a discussion but an actual discussion.

Please correct the language according to the comments made in the comments to authors.

Author Response

The manuscript “Dysfunctions of neutrophils in peripheral blood in children with cystic fibrosis” describes a prospective clinical study performed in 19 children with cystic fibrosis and 14 age-matched controls. The aim of the study was to evaluate the functional differences in peripheral blood neutrophils in children with cystic fibrosis compared to children without cystic fibrosis. The authors measured a large number of phenomena and stimulated the isolated neutrophils with several different inducers. This is of course of general interest to the field and an interesting alternative approach to understand what happens in cystic fibrosis. The experiments are carefully described and the text well written, with some minor errors in language.

There are some concerns with the manuscript which need to be addressed:

Major issues:

1: Figure 3: How can four values be enough to give statistical significance compared to three other groups? If correcting for multiple comparisons, there is no way this could be the case. Please use the correct statistical methods. On this same note, please indicate in all figure legends what statistical methods were used.

Thank you for this comment. We performed one-way ANOVA with the Bonferroni post-test, but unfortunately, we omitted to add this test for multiple comparisons in the sections Statistics. Now this is corrected, and as suggested, each used statistical test is added in Figure legends, including Supplementary figures. The limitation of the results when small groups are used is given in the Discussion ( last paragraph).

2: Page 4, line 152: Classification of lung damage: it would strengthen the grouping to include high resolution chest CT for evaluation of lung damage.

The chest CT was performed in some patients with a more severe form of the disease, but not in all patients. This notice was now added to the Material and Methods section

3: The discussion is too long. There is no way that the experiments performed can have such great impact that they merit such a long discussion. Please keep it short and to the point. It would also be of value if you briefly discussed the strengths and weaknesses of the study.

The discussion has been considerably shortened without significant omission of key points. Also, the main strength and weaknesses of the study have been outlined at the end of the Discussion.

4: Why did you not grow the Pseudomonas aeruginosa under mucoid conditions?

Your comment could be relevant to this study. Our explanation is as follows: Many in vitro studies related to Cystic fibrosis used conventional (a non-mucoid strain of P.aeruginosa). We decided to use this strain because although half of the children had chronic P.aeruginosa colonization most isolates were non-mucoid. In addition, mucoid P.aeruginosa is more relevant for airway neutrophils but not peripheral blood ones.

Minor issues:

1: The language needs some improvement and clarification:

Abstract, line 21: lesser examined. Please change to “less examined”.

Corrected

Page 2, line 90: Dysphynction. Please change to ”dysfunction”, if that is what you mean.

Corrected

Page 8, line 315: What does “Ninethin” mean? Nineteen?

Corrected (nineteen)

2: Please explain some statements made:

Page 3, line 110: Do you mean to say “2.1. Subsection”? The heading should read ”selection of participants” or something similar.

There was a mistake during the uploading of the manuscript. It is now named: Study participants

Page 4, line 159: What do you mean by “and its control”?

Corrected: disease control

Page 5, line 216: What is calcium ionophore (CaI)? It must have a name.

Corrected as calcium ionophore A23187

Page 5, line 242: What does “cultured as indicated for apoptosis” mean? Please describe how this was done.

Corrected: Neutrophils were cultured under identical conditions as described for the apoptosis assay.

Page 17, line 521: Apoptosis is the key biological function in the host’s immune response and inflammation. Please remove this statement as it is confusing.

Removed

Page 17, line 542: patients with gating (G551D) mutations. Either the patients have G551D or some other gating mutation (or both, on separate alleles). What do you mean?

The data you refer to was taken from reference 46 (Supplementary Table 1 of this paper). The gating G551D mutation was associated with the F508del  mutation, each on separate alleles. However, as you can see during revision and the discussion shortness this text has been deleted. Regarding this remark, we added the characteristics of our three children with compound heterozygous mutations (see Materials and Methods).

3: Please correct the numbering of the tables:

Table 3 should be table 1.

Sorry for this mistake which occurred by exchanging the order of the main sections. This table is now Supplementary Table 1.

Table 2 should be in section 2.2 and be called table 1.

Please make sure you have numbered the tables correctly and the text refers to the correct table. The demographics are not results but should be presented in the materials-section.

Yes, we have now supplementary Table 1 (previous Table 3). Table 1 has been moved to the section Materials and Methods (modified by the addition of parameters related to sex) with an explanation in the text that differences in any parameter are not statistically significant (required by another referee). In this context, subsection 2.1 is now the previous subsection 2.2 (subsequent subsections are renumbered). Previous Table 2 is modified by adding FEV1 and FVC parameters.

4: Please correct the text:

The acronym should be introduced the first time the substance or expression is mentioned, for example LPS. Then the acronym should be used and not explained again.

Checked in the whole text and corrected

Please remove the first sentences from the discussion (Authors should discuss the results and how they can be interpreted from the perspective of previous studies and of the working hypotheses. The findings and their implications should be discussed in the broadest context possible. Future research directions may also be highlighted.) This is not an instruction for how to write a discussion but an actual discussion.

Removed. Sorry for this mistake that occurred during the uploading of the manuscript

Comments on the Quality of English Language

Please correct the language according to the comments made in the comments to authors.

Checked in the whole text and corrected

Reviewer 3 Report

With interest, I read the manuscript biomedicines-2430571. I believe it is a valuable contribution based on a solid study.

I have only several, mostly minor remarks:

1.      The first table in the manuscript is … Table 3 (Table 3. Antibodies used in phenotypic analysis of neutrophils.). Anyway, it is too technical and should be moved to the supplements (with an appropriate number).

2.      Please, provide p-values to the data given in Table 1. Please, add sex to this table. Did the groups differ?

3.      Table 2. “Complex heterozygous” or, rather, “Compound heterozygous”?

4.      “Pseudomonas (P.) aeruginosa”, etc. always in italics (text, tables, figures).

5.      Please, optimize abbreviations strategy. For instance, “CF” is explained in the legend to each and every figure while less obvious abbreviations, e.g. “NE” re not at all. Please, make it also in the supplement. This applies also to the supplement.

6.      “Mann–Whitney test for quantitative variables” -> ““Mann–Whitney U test”.

7.      ”Graphical abstract would be welcome.

Minor editing of English language required

Author Response

With interest, I read the manuscript biomedicines-2430571. I believe it is a valuable contribution based on a solid study.

Thank you very much for this comment

I have only several, mostly minor remarks:

  1. The first table in the manuscript is … Table 3 (Table 3. Antibodies used in phenotypic analysis of neutrophils.). Anyway, it is too technical and should be moved to the supplements (with an appropriate number)

The table has been moved to the Supplementary material (now Supplementary Table 1)

  1. Please, provide p-values to the data given in Table 1. Please, add sex to this table. Did the groups differ?

Table 1 is now moved to Materials and Methods as suggested by another referee. We explained in the text that differences between CF and control groups did not differ in any parameter (p>0.05)

  1. Table 2. “Complex heterozygous” or, rather, “Compound heterozygous”?

Corrected (compound heterozygous)

  1. “Pseudomonas (P.) aeruginosa”, etc. always in italics (text, tables, figures).

Corrected

  1. Please, optimize abbreviations strategy. For instance, “CF” is explained in the legend to each and every figure while less obvious abbreviations, e.g. “NE” re not at all. Please, make it also in the supplement. This applies also to the supplement.

Corrected

  1. “Mann–Whitney test for quantitative variables” -> ““Mann–Whitney U test”.

Corrected

  1. ”Graphical abstract would be welcome.

We added now the graphical abstract

Comments on the Quality of English Language

Minor editing of English language required

Corrected

Round 2

Reviewer 1 Report

The authors took into account all my comments, I am completely satisfied. The article may be published in present form.

Reviewer 2 Report

The manuscript “Dysfunctions of neutrophils in peripheral blood of children 2
with cystic fibrosis
describes a prospective clinical study in children with cystic fibrosis and age-matched children without chronic airway conditions. The aim of the study was to evaluate the functional differences in peripheral blood neutrophils in children with cystic fibrosis compared to children without cystic fibrosis. This is of course of general interest to the field and an interesting alternative approach to understand what happens in cystic fibrosis. Generally, alveolar neutrophils are studied in the field and few studies have been conducted in peripheral blood neutrophils to evaluate their responses to various stimulations. The manuscript addresses this gap in knowledge. The experiments are carefully described, the methods correctly performed and wisely chosen to answer the questions asked. Cystic fibrosis is a rare disease, making it difficult to include large numbers of patients and in this context, the number of participants and their controls is sufficient. The conclusions are consistent with the evidence and they address the main question, which was to investigate responses in peripheral blood neutrophils in children with cystic fibrosis compared to age-matched controls. The references are appropriate and the tables and figures are useful and now numbered correctly. The graphical abstract is very informative.

The authors have responded appropriately to the reviewers´ comments and there are no major concerns. The editing process should include the following:

Line 137: R10070Q. I believe you mean R1070Q. CFTR does not contain over 10000 amino acids.

The English language is greatly improved but it does not hurt to do a second round of editing!